# Research on Dynamic Inertial Estimation Technology for Deck Deformation of Large Ships

**DOI:** 10.3390/s19194167

**Published:** 2019-09-25

**Authors:** Bo Ren, Tianjiao Li, Xiang Li

**Affiliations:** 1College of Equipment Engineering, Shenyang Ligong University, Shenyang 110159, China; Tianjiao912124@163.com; 2College of Information Science and Engineering, Shenyang Ligong University, Shenyang 110159, China; leexiang_lucky@163.com

**Keywords:** gyro sensor, deck deformation, dynamic inertial estimation, Kalman filter, wavelet

## Abstract

Many kinds of weapon systems and launching equipment on the deck of large ships are easily affected by deck deformation. In order to ensure the accuracy of weapon systems and the safety of taking off and landing of carrier aircraft, a dynamic estimation method combining the main inertial navigation systems (INS) and the sub-inertial navigation systems (SINS) is designed to estimate the curvature and torsion of any trajectory on the deck. Our contributions start from the fact that the area of concern extends from the fixed points to any trajectory on the deck. The dynamic filter algorithm of wavelet combined with Kalman filter is used to process the acquired data. The wavelet method is used to remove the outliers in the acquired data, and the Kalman filter effectively reduces the influence of white noise, so that the estimation accuracy is guaranteed. The simulation results clearly show that the deck deformation of large ships can be obtained accurately in real-time over the observed area which proved that this dynamic inertial measurement method is feasible in practical engineering application.

## 1. Introduction

Many sensors and weapon systems are installed on the deck of large ships, such as aircraft carriers and long-range survey ships. To ensure the proper work of these devices, they must be provided with accurate attitude [1]. These weapon systems will not work accurately when the ship’s deck is deformed [2,3]. Then the dynamic estimation of deck deformation of ships is carried out. The major problem to be solved is to estimate the real-time deformation of the deck, so the deck can be repaired or reinforced in time.

Among various measurement methods, such as optical, photogrammetric, and polarization methods, the most common method for estimating the deformation of a ship’s deck was the inertial measurement matching method [4,5]. This method began in the 1980s, and the United States, Great Britain, Germany, France, and other countries began to apply it to ship-borne attitude and heading datum successively. The Electrical Instrument Hall and the Electrical University in St. Petersburg, Russia have also carried out research in this area [6,7,8]. Its measurement principle is to install an Inertial Measurement Unit (IMU) [9] near the center of the ship’s deck as a reference point, and distribute the other one or more IMUs on the deck reasonably, then the output values of each IMU for determining the deck deformation are compared at that point [10,11]. Its advantages are high accuracy, simple structure, and easy operation, but the area coverage is small, comprising of only a few points.

Under the premise of ensuring the above advantages of the inertial measurement matching method, this paper proposes a dynamic inertial measurement method, which can extend the past single point measurement to the measurement of the entire observing trajectory.

## 2. Principle of Dynamic Inertial Measurement Method

The main inertial navigation system (INS) is installed in the center position of the deck of the ship as a benchmark, from which the position, velocity, acceleration, and angular motion are observed. Then a sub-inertial navigation system (SINS) slides on the deck to measure the deck angle motion generated by wave, constraint, and ship motion of the estimation trajectory several times. It means that the difference of angular motion between the SINS and the INS is the true angular deformation of the deck. At the same time, the real-time deformation position can be obtained from the SINS. This paper takes Russia’s “Kuznetsov” aircraft carrier as an example. Its landing deck is at *ε* = 7° to the ship’s central axis and is deflected to the port side [12]. The dotted line in Figure 1 is the estimation trajectory *op*.

The dynamic filtering algorithm of wavelet combined with Kalman filter is used on the two systems respectively, which can realize the noise reduction while removing the outliers [13]. The difference between the INS and the SINS angular motion filtering results is the angular motion of deck deformation. The curvature and torsion of deck deformation can be calculated according to differential geometry by using the angular motion results of deck deformation and the velocity of the SINS. And then the deformation is compared with the original design value to observe whether it exceeds the specified range. If the deformation of any part beyond the acceptable threshold is measured, the part will be located and repaired accordingly.

## 3. Estimation Models

### 3.1. Ship Benchmark Model

The INS selects the local horizontal coordinate system (also known as the geographic coordinate system) as the reference coordinate system (*L* system). In the local horizontal coordinate system, the three coordinate axes point to the east along the local latitude line (*e*-axis), the true north along the local meridian (*n*-axis), and the zenith along the normal of the local reference ellipsoid (*u*-axis) respectively. The dynamic model of the INS is as follows.

State equation of ship’s position is
(1)$${\left[\begin{array}{ccc} \dot{\phi } & \dot{\lambda } & \dot{h} \end{array}\right]}^{T}=D{\left[\begin{array}{ccc} v_{e} & v_{n} & v_{u} \end{array}\right]}^{T},$$ where ϕ, λ, and *h* are latitude, longitude, and altitude respectively. And **D** is given as below
(2)$$D=\left[\begin{array}{ccc} 0 & \frac{1}{M+h} & 0\\ \frac{1}{(N+h)cos\phi } & 0 & 0\\ 0 & 0 & 1 \end{array}\right],$$ where *M* is the radius of curvature of the earth meridian circle, and *N* is the radius of curvature of the prime vertical.

(3)$$\begin{array}{l} M\approx R_{e}(1-2\varepsilon +3\varepsilon sin^{2}\phi ),\\ N\approx R_{e}(1+\varepsilon sin^{2}\phi ). \end{array}$$

Earth radius $R_{e}=6,378,137$ and flat rate ε=1/298.257 are known in Equation (3).

State equation of ship’s velocity [3] is
(4)$$\left[\begin{array}{c} {\dot{v}}_{e}\\ {\dot{v}}_{n}\\ {\dot{v}}_{u} \end{array}\right]=\left[\begin{array}{c} f_{e}\\ f_{n}\\ f_{u} \end{array}\right]-\left(2\Omega_{iE}^{L}+\Omega_{EL}^{L}\right)\left[\begin{array}{c} v_{e}\\ v_{n}\\ v_{u} \end{array}\right]+g^{L},$$ where *i* represents the inertial coordinate system and *E* represents the ground-solid coordinate system. $\Omega_{iE}^{L}$ and $\Omega_{EL}^{L}$ are the anti-symmetric matrices of the angular velocity vectors $\omega_{iE}^{L}$ and $\omega_{EL}^{L}$ respectively, and $g^{L}$ is the gravity vector. Earth rotation angular velocity $\omega_{ie}=7.291158\times 10^{-5}\mathrm{rad}/s$ is known. The angular velocity vectors $\omega_{iE}^{L}$ and $\omega_{EL}^{L}$ are shown as below.

(5)$$\begin{array}{cc} \omega_{iE}^{L}=\left[\begin{array}{c} 0\\ \omega_{ie}cos\phi \\ \omega_{ie}sin\phi \end{array}\right] & \omega_{EL}^{L}=\left[\begin{array}{c} -\dot{\phi }\\ \dot{\lambda }cos\phi \\ \dot{\lambda }sin\phi \end{array}\right] \end{array}.$$

Therefore, the anti-symmetric matrices $\Omega_{iE}^{L}$ and $\Omega_{EL}^{L}$ become

(6)$$\begin{array}{cc} \Omega_{iE}^{L}=\left[\begin{array}{ccc} 0 & -\omega_{ie}sin\phi & \omega_{ie}cos\phi \\ \omega_{ie}sin\phi & 0 & 0\\ -\omega_{ie}cos\phi & 0 & 0 \end{array}\right] & \Omega_{EL}^{L}=\left[\begin{array}{ccc} 0 & -\dot{\lambda }sin\phi & \dot{\lambda }cos\phi \\ \dot{\lambda }sin\phi & 0 & \dot{\phi }\\ -\dot{\lambda }cos\phi & -\dot{\phi } & 0 \end{array}\right] \end{array}.$$

In the local horizontal coordinate system, the normal gravity vector has only the *u*-axis component, so
(7)$$g^{L}={\left[\begin{array}{ccc} 0 & 0 & -\gamma_{h} \end{array}\right]}^{T},$$ where
(8)$$\gamma_{h}=9.8\left(1+5.271\times 10^{-3}sin^{2}\phi +2.327\times 10^{-5}sin^{4}\phi \right)-3.086\times 10^{-6}h\left(m/s^{2}\right).$$

State equation of ship’s attitude angle is
(9)$${\left[\begin{array}{ccc} \dot{r} & \dot{p} & \dot{a} \end{array}\right]}^{T}={\left[\begin{array}{ccc} \omega_{r} & \omega_{p} & \omega_{a} \end{array}\right]}^{T},$$ where *r*, *p*, and *a* are the roll angle, the pitch angle and the azimuth angle respectively, and the angular velocities corresponding to them are $\omega_{r}$, $\omega_{p}$, and $\omega_{a}$.

State equation of ship’s attitude angular velocity is

(10)$$\left[\begin{array}{c} {\dot{\omega }}_{r}\\ {\dot{\omega }}_{p}\\ {\dot{\omega }}_{a} \end{array}\right]=\left[\begin{array}{ccc} 0 & -\frac{1}{M} & 0\\ \frac{1}{N} & 0 & 0\\ \frac{tan\dot{\phi }}{N} & 0 & 0 \end{array}\right]\left[\begin{array}{c} {\dot{v}}_{e}\\ {\dot{v}}_{n}\\ {\dot{v}}_{u} \end{array}\right]+\left[\begin{array}{c} 0\\ \omega_{ie}cos\dot{\phi }\\ \omega_{ie}sin\dot{\phi } \end{array}\right].$$

Therefore, the state vector of the INS is chosen as

(11)$$X_{L}={\left[\begin{array}{cccccccccccc} \phi & \lambda & h & v_{e} & v_{n} & v_{u} & r & p & a & \omega_{r} & \omega_{p} & \omega_{a} \end{array}\right]}^{T}.$$

And the state equation of the INS in the local horizontal coordinate system is
(12)$${\dot{X}}_{L}(t)=F_{L}(t)X_{L}(t)+W_{L}(t),$$ where ***F****_L_* is a 12 × 12 dimensional state transition matrix, the 12-dimensional state noise vector is

(13)$$W_{L}={\left[\begin{array}{cccccccccccc} 0 & 0 & 0 & 0 & 0 & -\gamma_{h} & 0 & 0 & 0 & 0 & \omega_{ie}cos\dot{\phi } & \omega_{ie}sin\dot{\phi } \end{array}\right]}^{T}.$$

The position and attitude angular velocity of the INS are taken as observation information, and the observation vector of the INS is
(14)$$Z_{L}(t)={\left[\begin{array}{cccccc} \phi & \lambda & h & \omega_{r} & \omega_{p} & \omega_{a} \end{array}\right]}^{T}$$ so the observation equation of the INS is
(15)$$Z_{L}(t)=H_{L}(t)X_{L}(t)+V_{L}(t),$$ where the observation matrix consists of a 3 × 3 dimensional unit matrix $I_{3\times 3}$ and 3 × 9 dimensional zero matrix $0_{3\times 9}$.

(16)$$H_{L}=\left[\begin{array}{cc} I_{3\times 3} & 0_{3\times 9}\\ 0_{3\times 9} & I_{3\times 3} \end{array}\right].$$

The observation noise vector is a 6-dimensional zero-mean white noise vector [14], which is

(17)$$\left\{\begin{array}{l} E\left[V_{L}(t)\right]=0\\ E\left[V_{L}(t)V_{L}^{T}(\tau )\right]=R(t)\delta (t-\tau ) \end{array}\right.,$$

So the continuous space state model of the INS is established. Next is the establishment of the SINS model.

### 3.2. Sliding Estimation Model

As shown in Figure 1, the SINS slides uniformly on the estimation trajectory. Its kinematic model in the local horizontal coordinate system is as follows.

The position state equation of the SINS on the estimation trajectory is given as

(18)$$\dot{s}={\left[\begin{array}{ccc} {\dot{s}}_{x} & {\dot{s}}_{y} & {\dot{s}}_{z} \end{array}\right]}^{T}={\left[\begin{array}{ccc} v_{x} & v_{y} & v_{z} \end{array}\right]}^{T}.$$

The velocity state equation of the SINS on the estimation trajectory is defined as

(19)$$\dot{v}={\left[\begin{array}{ccc} {\dot{v}}_{x} & {\dot{v}}_{y} & {\dot{v}}_{z} \end{array}\right]}^{T}=0.$$

The deflection angle of the deck caused by the motion of the ship, wave, and constraint in the ocean is at least a second-order model, and the angular motion of the estimation system is approximated as a second-order Markov stochastic process [15]. It should be emphasized that this is just one model from among many which could be adopted. The parameters chosen is complex enough to represent the true situation with a fair level of fidelity, yet simple enough to illustrate the development of the Kalman filter. So the angular velocity state equation of the SINS on the estimation trajectory is defined as [16]
(20)$$\dot{\omega }=\left[\begin{array}{c} {\dot{\omega }}_{x}\\ {\dot{\omega }}_{y}\\ {\dot{\omega }}_{z} \end{array}\right]=\left[\begin{array}{ccc} -\beta_{1}^{2} & 0 & 0\\ 0 & -\beta_{2}^{2} & 0\\ 0 & 0 & -\beta_{3}^{2} \end{array}\right]\left[\begin{array}{c} \lambda_{x}\\ \lambda_{y}\\ \lambda_{z} \end{array}\right]+\left[\begin{array}{ccc} -2\beta_{1} & 0 & 0\\ 0 & -2\beta_{2} & 0\\ 0 & 0 & -2\beta_{3} \end{array}\right]\left[\begin{array}{c} \omega_{x}\\ \omega_{y}\\ \omega_{z} \end{array}\right],$$ where *β*_1_, *β*_2_, and *β*_3_ are the inverse correlation times of the corresponding stochastic processes, and *λ**_x_*, *λ**_y_*, and *λ**_z_* are three-axis attitude angles of the SINS. The attitude angles are obtained by the attitude matrix ***R****_S_*. The coordinate transformation matrix of the SINS carrier coordinate system to the local horizontal coordinate system is given as [17]

(21)$$\begin{array}{ll} R_{S} & =\left[\begin{array}{ccc} cos\lambda_{z}cos\lambda_{x}-sin\lambda_{z}sin\lambda_{y}sin\lambda_{x} & -sin\lambda_{z}cos\lambda_{y} & cos\lambda_{y}sin\lambda_{x}+sin\lambda_{z}sin\lambda_{y}cos\lambda_{x}\\ sin\lambda_{z}cos\lambda_{x}+sin\lambda_{y}cos\lambda_{z}sin\lambda_{x} & cos\lambda_{z}cos\lambda_{y} & sin\lambda_{z}sin\lambda_{x}-cos\lambda_{z}sin\lambda_{y}cos\lambda_{x}\\ -cos\lambda_{y}sin\lambda_{x} & sin\lambda_{y} & cos\lambda_{y}cos\lambda_{x} \end{array}\right]\\ & =\left[\begin{array}{ccc} R_{11} & R_{12} & R_{13}\\ R_{21} & R_{22} & R_{23}\\ R_{31} & R_{32} & R_{33} \end{array}\right]. \end{array}$$

The coordinate transformation matrix state equation of the SINS on the estimation trajectory is

(22)$${\dot{R}}_{S}=R_{S}\Omega =R_{S}\left[\begin{array}{ccc} 0 & -\omega_{z} & \omega_{y}\\ \omega_{z} & 0 & -\omega_{x}\\ -\omega_{y} & \omega_{x} & 0 \end{array}\right].$$

And the *λ**_x_*, *λ**_y_*, and *λ**_z_* are obtained by the following equation

(23)$$\left\{\begin{array}{l} \lambda_{x}=arctan(-\frac{R_{31}}{R_{33}})\\ \lambda_{y}=arcsinR_{32}\\ \lambda_{z}=arctan(\frac{R_{12}}{R_{22}}) \end{array}\right..$$

Therefore, the state vector of the SINS on the estimation trajectory is chosen as

(24)$$X_{S}={\left[\begin{array}{cccc} s & v & \omega & R \end{array}\right]}^{T}.$$

And the state space model of the SINS on the estimation trajectory is
(25)$${\dot{X}}_{S}=F_{S}X_{S}+W_{S},$$ where ***F****_S_* is a state transition matrix, and the state noise vector is a 12-dimensional zero-mean white noise vector, which is
(26)$$\left\{\begin{array}{l} E\left[W_{S}(t)\right]=0\\ E\left[W_{S}(t)W_{S}^{T}(\tau )\right]=Q(t)\delta (t-\tau ) \end{array}\right..$$

The position and attitude angular velocity of the SINS are taken as observation information, and the observation vector of the SINS is
(27)$$Z_{S}(t)={\left[\begin{array}{cccccc} s_{x} & s_{y} & s_{z} & \omega_{x} & \omega_{y} & \omega_{z} \end{array}\right]}^{T},$$ so the observation equation of the SINS is
(28)$$Z_{S}(t)=H_{S}(t)X_{S}(t)+V_{S}(t),$$ where the observation matrix consists of 3 × 3 dimensional unit matrix $I_{3\times 3}$ and 3 × 3 dimensional zero matrix $0_{3\times 3}$.

(29)$$H_{S}=\left[\begin{array}{cccc} I_{3\times 3} & 0_{3\times 3} & 0_{3\times 3} & 0_{3\times 3}\\ 0_{3\times 3} & 0_{3\times 3} & I_{3\times 3} & 0_{3\times 3} \end{array}\right].$$

The observation noise vector is a 6-dimensional zero-mean white noise vector, which is

(30)$$\left\{\begin{array}{l} E\left[V_{S}(t)\right]=0\\ E\left[V_{S}(t)V_{S}^{T}(\tau )\right]=R(t)\delta (t-\tau ) \end{array}\right..$$

The above is the SINS sliding estimation model.

Based on the differential geometry knowledge, the curvature and torsion parameters of the estimation trajectory can be calculated by using the attitude information output from the above-mentioned mechanical arrangement and modified by filtering [18,19]. The vertical curvature is
(31)$$\kappa_{V}=\frac{d\theta_{y}}{ds}=\frac{d(\lambda_{y}-p)}{ds}=\frac{\omega_{y}-\omega_{p}}{v}=\frac{\omega_{y}-\omega_{p}}{\sqrt{v_{x}^{2}+v_{y}^{2}+v_{z}^{2}}},$$ where *v* is the velocity of the SINS, and the horizontal curvature is

(32)$$\kappa_{H}=\frac{d\theta_{z}}{ds}=\frac{d(\lambda_{z}-a)}{ds}=\frac{\omega_{z}-\omega_{a}}{v}=\frac{\omega_{z}-\omega_{a}}{\sqrt{v_{x}^{2}+v_{y}^{2}+v_{z}^{2}}}$$

And the torsion is

(33)$$\tau =\frac{d\theta_{x}}{ds}=\frac{d(\lambda_{x}-r)}{ds}=\frac{\omega_{x}-\omega_{r}}{v}=\frac{\omega_{x}-\omega_{r}}{\sqrt{v_{x}^{2}+v_{y}^{2}+v_{z}^{2}}}.$$

Before the curvature and torsion calculation, the filtering algorithm of wavelet combined with Kalman filter is applied to the INS and SINS models.

## 4. Dynamic Filtering Algorithm

Wavelet method is popular for being treated as the mathematical microscope for analyzing signals. The outliers are the mutation jump point in the observation sequence, which belongs to the detail part of the observed signal. And wavelet is the most appropriate method to deal with it. The outliers, noise, etc. in the signal are often high-frequency components, which are identified by the wavelet function and set to zero then rebuild to achieve the purpose of removing them [20,21]. Kalman filter is a real-time recursive algorithm that reduces the effects of white noise. The dynamic algorithm of wavelet combined with Kalman filter is summarized into the following two steps.

The first step is to choose the wavelet basis function. Considering the boundary, a 3rd-order Daubechies wavelet with a filter length of 6 and a support length of 5 is selected as the wavelet basis function, and shape of the function is similar to the shape of outliers as shown in Figure 2.

The second step is to filter the observed signal. The standard discrete Kalman filter algorithm is summarized as follows [22,23]

Prediction: $\hat{X}(\left.k+1\right|k)=\Phi (\left.k+1\right|k)\hat{X}(\left.k\right|k)$Correction: $\hat{X}(\left.k+1\right|k+1)=\hat{X}(\left.k+1\right|k)+K(k+1){\left[Z(k+1)-H(k+1)\hat{X}(\left.k+1\right|k)\right]}^{}$Kalman gain matrix: $K(k+1)=P(\left.k+1\right|k)H^{T}(k+1){\left[H(k+1)P(\left.k+1\right|k)H^{T}(k+1)+R(k+1)\right]}^{-1}$Prediction error variance matrix: $P(\left.k+1\right|k)=\Phi (\left.k+1\right|k)P(\left.k\right|k)\Phi^{T}(\left.k+1\right|k)+Q(k)$Correction error variance matrix: $P(\left.k+1\right|k+1)=P(\left.k+1\right|k)-K(k+1)H(k+1)P(\left.k+1\right|k)$

If the amount of signal points is less than six, Kalman filter can be applied directly. When the number of signal points is greater than or equal to six, the signal will be divided into six points for wavelet decomposition and reconstruction as Equation (34).
(34)$$\begin{array}{l} Z_{m}(i-1,k)=\sum_{n}h(2k-n)Z(i,n)=\sum_{n=0}^{L-1}h(n)Z(i,2k-n)\\ Z_{d}(i-1,k)=\sum_{n}g(2k-n)Z(i,n)=\sum_{n=0}^{L-1}g(n)Z(i,2k-n)\\ Z(i,k)=\sum_{n}h(2k-n)Z_{m}(i-1,k)+\sum_{n}g(2k-n)Z_{d}(i-1,k) \end{array}$$ where *L* = 6, *h* is scaling function, *g* is wavelet function, ***Z****_m_* is smooth approximation and ***Z****_d_* is detail signal.

The last point of the processing result is reserved and sent to Kalman filter, then the signal point moves in turn. The above process is repeated until the measurement ends.

## 5. Deck Deformation Measurement Simulation

### 5.1. Parameter Setting

The “Kuznetsov” aircraft carrier was chosen as the simulation object. First, the initial latitude and longitude of the ship were 110° and 30° respectively. The ship sailed eastward at a velocity of 18 kn. The angular velocity of the ship was assumed to be ${\left[\begin{array}{ccc} \omega_{r} & \omega_{p} & \omega_{a} \end{array}\right]}^{T}={\left[\begin{array}{ccc} 0 & 0 & 0 \end{array}\right]}^{T}$. Second, the SINS chose an inertial navigation system with an accuracy of 10n mile/h, and its equivalent gyro drift was 0.1°/h, that is, $\omega ={\left[\begin{array}{ccc} 0.1^{\circ }/h & 0.1^{\circ }/h & 0.1^{\circ }/h \end{array}\right]}^{T}$. The initial attitude angle was selected as ${\left[\begin{array}{ccc} \lambda_{x} & \lambda_{y} & \lambda_{z} \end{array}\right]}^{T}={\left[\begin{array}{ccc} 5^{\prime } & 5^{\prime } & 20^{\prime } \end{array}\right]}^{T}$. It assumed that the inverse correlation times of the corresponding stochastic processes *β*_1_, *β*_2_, *β*_3_ were 0.15, 0.12, and 0.10 respectively. And the SINS moved uniformly along the estimation trajectory at a velocity of 5 m/s, the estimation trajectory was 100 m long, the estimation time was 20 s.

### 5.2. Results and Discussion

#### 5.2.1. Simulation Results of the Ship Benchmark Model

The state values, observation values, Kalman filter values, and db3 wavelet combined with Kalman filter values in the absence of noise of the angular velocity of the INS are shown in Figure 3 below.

Due to the influences of random noise, the angular velocity observation (as shown by the red line in Figure 3) contains errors. In order to improve the estimation accuracy, the Kalman filter algorithm is used in the dynamic measurement process (as shown by the blue line in Figure 3) alone. The recursive estimation is performed, and the results show that the Kalman filter has a substantial noise reduction effect. In order to pursue higher observation accuracy, the db3 wavelet combined with the Kalman filter algorithm (as shown by the green line in Figure 3) is used to dynamically process the system. The comparison in Figure 3 is obvious. The db3 wavelet combined with Kalman filter results is closer to the simulated true values (as shown by the black line in Figure 3), which is better than using only the Kalman filter. This combination algorithm is particularly advantageous in removing outliers.

#### 5.2.2. Simulation Results of the Sliding Estimation Model

The true angular velocity values of the SINS without noise (left side of the Figure 4) are compared with the angular velocity state values, observation values, Kalman filter values, and wavelet combined Kalman filter values of the SINS with noise (right side of the Figure 4).

It can be seen from Figure 4 that the wavelet combined with Kalman filter values are closest to the true values. The filtering effect can also be analyzed from the perspective of the frequency. The spectrum of the observation of the angular velocity is shown on the left side of the Figure 5. The spectrum of the angular velocity combined with Kalman filter values is shown on the right side of the Figure 5.

The simulated true angular velocity of the SINS estimation system is a low frequency signal, and the interference component such as noise are some high frequency signals. The results before and after filtering are compared. It can be seen that the high frequency component of the signal is suppressed, and the low frequency signal is preserved after filtering. The main frequency of the angular velocity 0.09766 Hz remains unchanged as shown in Figure 5. The high frequency component in Figure 5f still exists, but does not accumulate in a large amount at a certain frequency, and it can be inferred that the filtering is effective.

The mean and RMS of the angular velocity before and after filtering are shown in Table 1.

As can be seen from Table 1, the precision after filtering is an order of magnitude higher than before. Besides the mean value of angular velocity after filtering is close to its true value, which means that the accuracy of the measurement has also been improved. The data in Table 2 were published by Sameh Nassar and Naser El-Sheimy [24] using solely wavelet, and the accuracy is also improved. The difference is that this paper uses the wavelet combined with the Kalman filter algorithm.

The results show that the dynamic filtering algorithm of db3 wavelet combined with Kalman filter is more precise than solely the Kalman algorithm or wavelet method.

#### 5.2.3. Curvature and Torsion of the Deck

The curvature and torsion on the estimation trajectory are shown in the Figure 6.

Since the filtering algorithm of wavelet combined with Kalman filter has poor filtering ability to the initial values, the initial values of curvature and torsion in Figure 6 fluctuate greatly. The vertical curvature of Figure 6a gradually decreases with the change of the estimation position, the horizontal curvature of Figure 6b remains stable. The torsion of Figure 6c also becomes smaller as the estimation position is closer to the center of the deck, and the negative sign represents left-handed rotation. It shows that the deformation of the deck edge is obvious, while the deformation of the deck center is small, which is consistent with the actual situation.

The curvature and torsion parameters obtained in Figure 6 are compared with the original design values, and the part exceeding the design deformation range is found, and the specific position of the estimation track is located, as shown in Figure 7.

The specific location of the SINS is given in Figure 7. It is clearly shown in Figure 7 that the measurement trajectory is relatively smooth without outliers, which is in line with the actual situation, indicating that the wavelet combined with Kalman filter algorithm has significant advantages in improving measurement accuracy. As shown in Figure 6, the first 3 s of torsion and curvature exceed the threshold, i.e., the first 6 meters are the potential danger zone. A partial enlargement of the *xoy* plane is drawn. Accordingly, the potential danger zones on the deck are repaired or reinforced to ensure that the accuracy of various equipment on the ship is not affected when the ship is sailing.

In summary, the appropriate scenario design yields ideal results. It is suitable to describe the deflection of the deck with a second-order Markov model, which verifies the effectiveness of the measurement method. Furthermore, it can be inferred that if the deck deformation is described by a more accurate model, the simulation results will be more accurate.

## 6. Conclusions

In this paper, the deflection of the deck is measured by a fixed INS and a sliding SINS, instead of the previous inertial measurement matching method. Additionally, a second-order Markov model is used to simulate the *op* estimation trajectory with a length of 100 m on the landing deck of the “Kuznetsov”. The results show that the first 6 m are potential danger zone. Additionally, the data obtained by wavelet combined with Kalman filter algorithm indicate that the noise and outliers can be removed without changing the main frequency of 0.09766 Hz, and the accuracy is improved by an order of magnitude. Therefore, three remarkable advantages of the proposed method are high accuracy, dynamic and widespread measurement range. However, the proposed method requires extremely high accuracy of the sensors and it is necessary to ensure that the SINS closely fits the deck during the measurement process.

In conclusion, this dynamic measurement method is promising in experiment and engineering practice. In the future, more complicated deck situations should be measured.

## Figures and Tables

**Figure 1 sensors-19-04167-f001:**
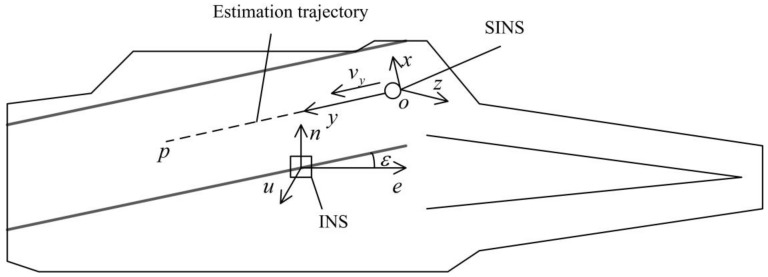
Principle diagram of dynamic inertial measurement method.

**Figure 2 sensors-19-04167-f002:**
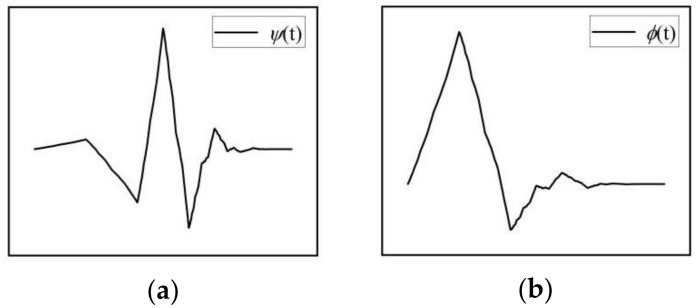
3rd-order Daubechies wavelet. (**a**) Wavelet function; (**b**) Scaling function.

**Figure 3 sensors-19-04167-f003:**
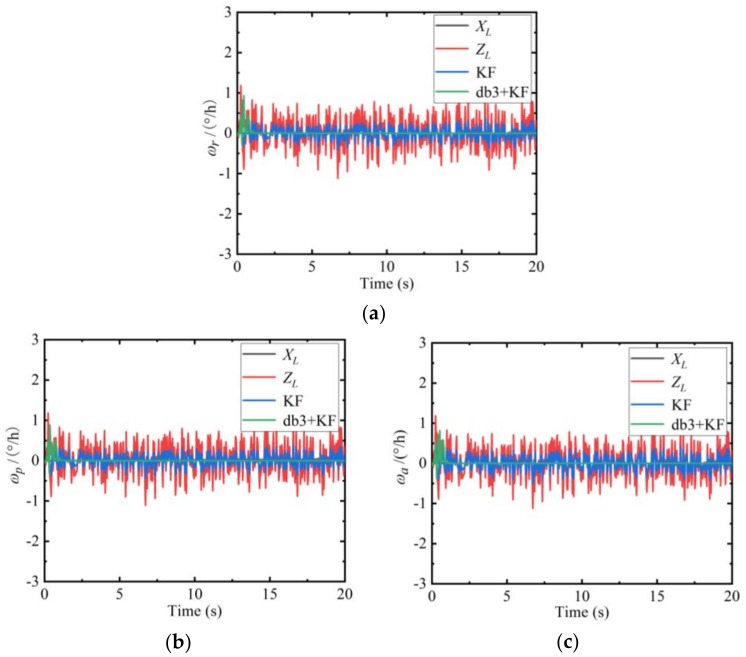
Comparison of angular velocity filtering results of the inertial navigation systems (INS). (**a**) $\omega_{r}$ of the INS; (**b**) $\omega_{p}$ of the INS; (**c**) $\omega_{a}$ of the INS.

**Figure 4 sensors-19-04167-f004:**
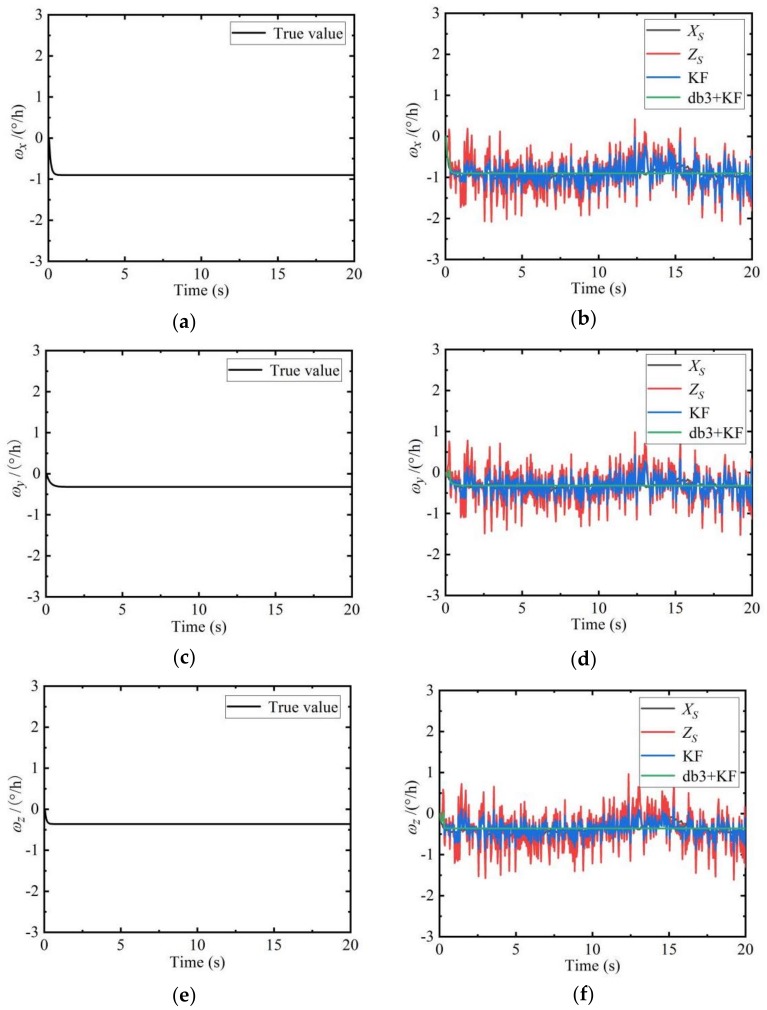
Comparison of angular velocity filtering results of the sub-inertial navigation systems (SINS). (**a**) Noiseless $\omega_{x}$ of the SINS; (**b**) Noisy $\omega_{x}$ of the SINS; (**c**) Noiseless $\omega_{y}$ of the SINS; (**d**) Noiseless $\omega_{y}$ of the SINS; (**e**) Noisy $\omega_{z}$ of the SINS; (**f**) Noiseless $\omega_{z}$ of the SINS.

**Figure 5 sensors-19-04167-f005:**
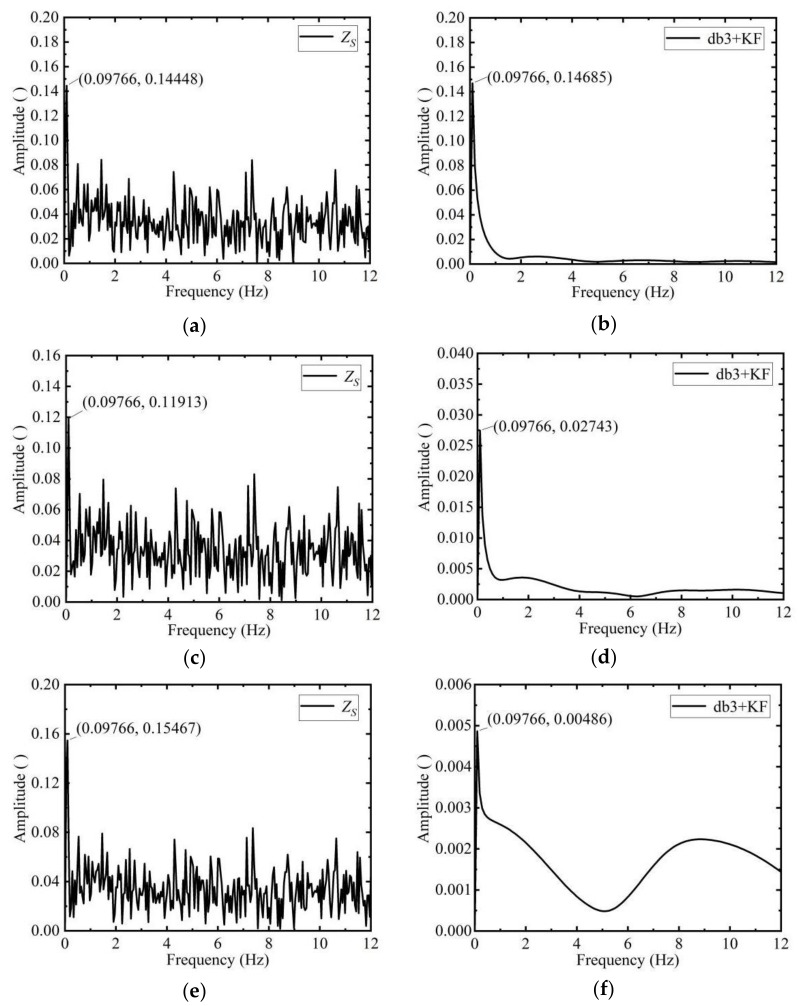
Spectrum analysis before and after filtering of the SINS angular velocity. (**a**) Spectrogram of observation values of the $\omega_{x}$; (**b**) Spectrogram of filter values of the $\omega_{x}$; (**c**) Spectrogram of observation values of the $\omega_{y}$; (**d**) Spectrogram of filter values of the $\omega_{y}$; (**e**) Spectrogram of observation values of the $\omega_{z}$; (**f**) Spectrogram of filter values of the $\omega_{z}$.

**Figure 6 sensors-19-04167-f006:**
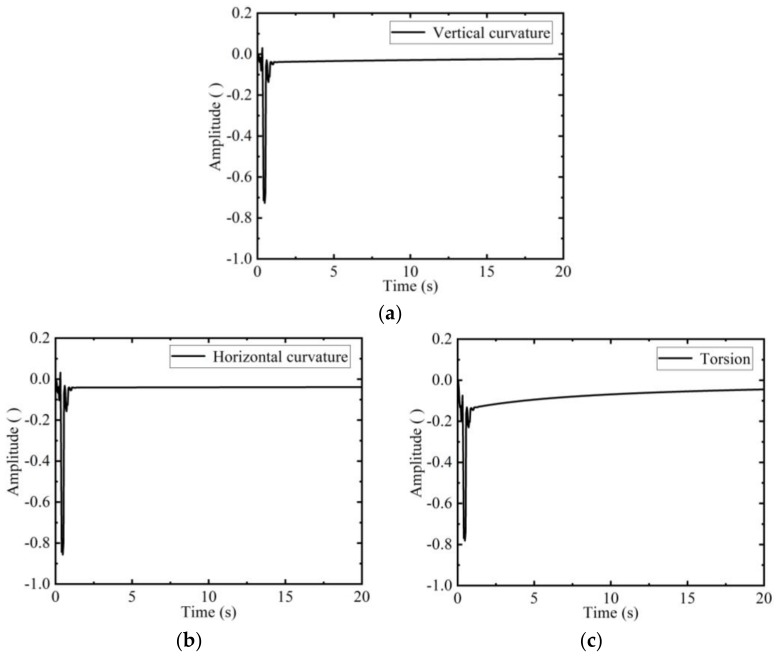
Curvature and torsion of the estimation trajectory: (**a**) Vertical curvature of the deck; (**b**) Horizontal curvature of the deck; (**c**) Torsion of the deck.

**Figure 7 sensors-19-04167-f007:**
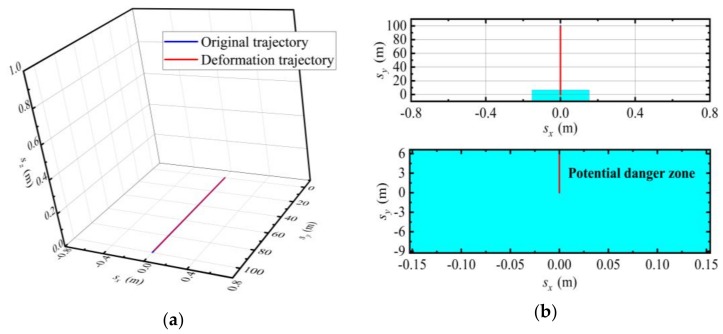
Sliding position of the estimation trajectory: (**a**) Original trajectory and deformation trajectory; (**b**) Partial enlargement of the *xoy* plane (Potential danger zone).

**Table 1 sensors-19-04167-t001:** Angular velocity errors of the SINS before and after filtering.

Axial	Parameter	Original Data	Filter Data (Wavelet Combined with Kalman Filter)
*x*	Mean	0.9180	0.3812
	RMS	1.0157	0.4036
*y*	Mean	0.3271	0.1460
	RMS	0.5328	0.1487
*z*	Mean	0.3874	0.1974
	RMS	0.5833	0.1981

**Table 2 sensors-19-04167-t002:** INS position errors before and after wavelet de-noising of inertial data.

Parameter	Original Data	Filter Data (Wavelet)
Mean	1.76	0.64
RMS	1.98	0.76

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
