# Peer review of "Research on Dynamic Inertial Estimation Technology for Deck Deformation of Large Ships"

_sensors, 2019, doi:10.3390/s19194167_

Round 1

Reviewer 1 Report

Dear author

The work present a series of simulation to confirm reliability of Inertial Measurement Unit for deformation detection of large ships, which is interesting and can be advantageous in field of marine safety and design.

Despite of the positive aspect, readability of this work is quite low due to the most important part of the explanation is not presented, i.e. setting of the simulation. Authors just wrote: The "Kuznetsov" aircraft carrier is chosen as the simulation object. The measurement accuracy 163 of the INS and the SINS are both 10n mile/h. The initial conditions for the simulation of the INS 164 model is that the ship sails eastward at a velocity of 18 kn in the ocean. And the SINS moves 165 uniformly along the estimation trajectory at a velocity of 5 m/s , the estimation trajectory is 100 m 166 long, so the estimation time is 20 s., to explain various things in simulation.

Authors is highly recommended to properly explain these items:

1. Ship model

2. Definition of boundary condition (location of sensors, wave consideration and constraint)

3. Scenario design which will be discussed in Results and discussion.

Moreover, setting and configuration have to be in separate section. Do not mix it with Results Analysis (or the authors mean Results and Discussions).

If these parts are not explained well, the information in this work will be failed to be reproduced by future reader.

Author Response

Dear reviewer,

Thank you very much for reviewing our manuscript thoroughly, which reminds me of a line in the American drama "The Big Bang Theory", which has been revisited in recent days: "If the reader carries out the same experiment according to your article, but can't get the same conclusions, then this article is meaningless." We highly value your opinions and agree with it. And we have made the following modifications.

First, the introduction is supplemented, and a variety of methods for measuring deck deformation are listed, such as optical, photogrammetric, polarization and inertial matching method. Besides, two references have been added ([4] Liu, H.B.; Sun, C.; Zhang, Y.Q.; Liu, X.M.; Liu, J.B.; Zhang, X.H.; Yu, Q.F. Hull deformation measurement for spacecraft TT&C ship by Photogrammetry. Science China Technological Sciences 201558, 1339–1347. [5]Yuan, E.K.; Yang, G.L. High-Accuracy Modeling of Ship Deformation Based on Inertial Measuring Method. Advanced Materials Research 2013760-762, 1227–1232.).

Second, for your comments, our modifications are as follows.

    1. Ship model

The model in the manuscript is divided into two parts, namely the motion of the ship (the INS) and the deformation of the deck (the SINS). The angular motion of the INS is subtracted from the angular motion information obtained by the SINS, thus more accurate deck deformation can be obtained. A more detailed description of the “Principle of Dynamic Inertial Measurement Method” in the manuscript is given.

Since this paper mainly studies the deformation and deformation position of the deck, the ship model is not considered too much, and only the angular motion is constrained.

    2. Definition of boundary condition (location of sensors, wave consideration and constraint)

To be honest, in order to simplify the calculation, this problem is an equivalent replacement, using a second-order Markov process to cover all conditions, including ship motion, wave effects, and constraints (Reference [16]). However, it does not affect the experimental results. Markov model is a commonly and widely used model for deflection problems. And this is just one model from among many which could be adopted. The parameters chosen is complex enough to represent the true situation with a fair level of fidelity, yet simple enough to illustrate the development of the Kalman filter.

    3. Scenario design which will be discussed in Results and Discussion.

The discussion is supplemented in section '5.2 Results and discussion' of the article as follows.

It is clearly shown in Figure 7 that the measurement trajectory is relatively smooth without outliers, which is in line with the actual situation, indicating that the wavelet combined with Kalman filter algorithm has significant advantages in improving measurement accuracy. As shown in Figure 6, the first 3 seconds of torsion and curvature exceed the threshold, i.e., about the first 6 meters are the potential danger zone. And a partial enlargement of the xoy plane is drawn. Accordingly, the potential danger zone on the deck are repaired or reinforced to ensure that the accuracy of various equipment on the ship is not affected when the ship is sailing.

In summary, the appropriate scenario design yields ideal results. It is suitable to describe the deflection of the deck with a second-order Markov model, which verifies the effectiveness of the measurement method. Furthermore, it can be inferred that if the deck deformation is described by a more accurate model, the simulation results will be more accurate.

Moreover, the parameter setting section are added separately in the manuscript.

"5.1. Parameter setting

The "Kuznetsov" aircraft carrier was chosen as the simulation object. First, the initial latitude and longitude of the ship were 110° and 30° respectively. And the ship sailed eastward at a velocity of 18 kn. The angular velocity of the ship was assumed to be [ωr,ωp,ωa]T=[0,0,0]T. Second, the SINS chose an inertial navigation system with an accuracy of 10n mile/h, and its equivalent gyro drift was 0.1°/h, that is, ω=[0.1°/h,0.1°/h,0.1°/h]T. The initial attitude angle was selected as [λx,λy,λz]T=[5',5',20']T. It assumed that the inverse correlation times of the corresponding stochastic processes β1β2β3 were 0.15, 0.12 and 0.10 respectively. And the SINS moved uniformly along the estimation trajectory at a velocity of 5 m/s, the estimation trajectory was 100 m long, so the estimation time was 20 s."

Finally, the conclusion is improved. "6. Conclusions

In this paper, the deflection of the deck is measured by a fixed INS and a sliding SINS, instead of the previous inertial measurement matching method. And a second-order Markov model is used to simulate the op estimation trajectory with a length of 100 m on the landing deck of the "Kuznetsov". The results show that the first 6 m are potential danger zone. Additionally, the data obtained by wavelet combined with Kalman filter algorithm indicate that the noise and outliers can be removed without changing the main frequency of 0.09766 Hz, and the accuracy is improved by an order of magnitude. Therefore, three remarkable advantages of the proposed method are high accuracy, dynamic and widespread measurement range. However, the proposed method requires extremely high accuracy of the sensors and it is necessary to ensure that the SINS closely fits the deck during the measurement process.

In conclusion, this dynamic measurement method is promising in experiment and engineering practice. In the future, more complicated deck situations should be measured."

Thank you again for reviewing our manuscript, your suggestions are very helpful, and we really look forward to your agreement to publish this article.

Sincerely,

Authors

Reviewer 2 Report

A very interesting article using elements of simulation tests and real measurements. However, figure 7 should be improved, as there is no possibility to assess the correctness of test results. Figure 7 is important for evaluating the entire measurement methodology. In the conclusions, please indicate the benefits and possible disadvantages of using this solution

Round 2

Reviewer 1 Report

Previous comments have been clearly addressed in the current rebuttal and revised manuscript.

Reviewer declares that the manuscript is now acceptable for publication.